# Utility of the Comprehensive Trail Making Test in the Assessment of Mild Cognitive Impairment in Older Patients

**DOI:** 10.3390/geriatrics8060108

**Published:** 2023-10-31

**Authors:** Adam Bednorz, Dorota Religa

**Affiliations:** 1John Paul II Geriatric Hospital, 40-353 Katowice, Poland; adam.bednorz@humanitas.edu.pl; 2Institute of Psychology, Humanitas Academy, 41-200 Sosnowiec, Poland; 3Division of Clinical Geriatrics, Department of Neurobiology, Care Sciences and Society, Karolinska Institutet, 171 77 Stockholm, Sweden; 4Theme Inflammation and Aging, Karolinska University Hospital, 141 86 Huddinge, Sweden

**Keywords:** cognitive impairment, executive functions, neuropsychological assessment

## Abstract

Introduction. The purpose of this study is to determine the usefulness of the CTMT (Comprehensive Trail Making Test) in diagnosing mild cognitive impairment in older patients. The test is used to assess executive functions, of which impairment is already observed in the early stages of the neurodegenerative process. Materials and Methods. The study includes 98 patients of a geriatric ward assigned to 2 groups of 49 patients each: patients diagnosed with a mild cognitive impairment and patients without a cognitive impairment, constituting the control group (group K). A set of screening tests was used in the initial study: the MMSE (Mini-Mental State Examination), MoCA (Montreal Cognitive Assessment), and CDT (Clock Drawing Test), GDS (Geriatric Depression Scale). The second study included the performance of the CTMT; the performance indicator was the time of performance. Results. Statistically significant differences are obtained between patients with mild cognitive impairments and those in cognitive normality in the performance of the CTMT test (*p* < 0.01). Patients with MCIs took longer to complete all trails of the test. To identify cognitive impairment, cutoff points were proposed for the CTMT total score and the other test trails. The CTMT overall score and CTMT 5 scored the highest AUCs (CTMT overall score = 0.77, CTMT Trail 5 = 0.80). Conclusions. The Comprehensive Trail Making Test may be useful in diagnosing mild cognitive impairment as a complementary screening tool.

## 1. Introduction

Dementia is preceded by a preclinical period of several years, in which the pathological mechanisms causing an increase in the central nervous system content of β-amyloid42, β-amyloid40, and tau protein are already activated. Developments in the study of biomarkers of AD pathology make it possible to diagnose the disease even 20 years earlier, before the onset of a specific clinical phenotype [1,2,3]. The recognition of AD biomarkers has made it possible to define three stages of disease development: (1) the early stage without clinical symptoms, (2) the middle stage of mild cognitive impairment (MCI), and (3) the final stage characterized by developed dementia [4,5]. When analyzing the data from the clinical trials and those verifying the efficacy of therapeutic interventions, it can be seen that the researchers’ attention has shifted at present from dementia to MCI [6].

The modern diagnostic criteria for MCI focus on expanding the cognitive impairment classification scheme to include other cognitive functions in addition to memory. Increasingly, a number of combinations of deficits within MCI are being advocated: amnestic cognitive impairment (deficits only in episodic memory), so-called multi-domain amnestic MCI, which includes an amnestic deficit and a deficit in another cognitive function, such as executive functions, and non-amnestic cognitive impairment (i.e., the impairment of a function other than episodic memory) [7]. Traditionally, an amnestic MCI is the typical prodromal stage of AD-induced dementia, but other phenotypes can also lead to this type of dementia, such as logopenic aphasia, posterior cortical atrophy (also known as visual variant), or AD presentation with frontal lobe and executive functions impairment. A key feature of this account is that not all MCIs are early AD [8,9]. The criteria for the pre-dementia stage have been developed for other forms of dementia, such as vascular dementia and dementia with Lewy bodies [10,11]. MCIs are common in older populations and its prevalence increases with age and lower levels of education. Individuals with MCIs are at greater risk of progression to dementia than age-matched controls. However, individuals diagnosed with MCIs may progress to dementia, may remain at MCI levels, or may improve the cognitive performance [9].

Studies have indicated that the rate of damage to individual cognitive functions is uneven [12]. In the longitudinal studies of MCI, a faster progression was observed in executive functions than in memory [13]. Expert recommendations suggest the need to include tests specific to impairments in episodic memory, social cognition, and other cognitive functions (according to the DSM-5 guidelines) to increase the sensitivity of the battery for typical AD, atypical AD, and for the behavioral variant of frontotemporal dementia [14]. Invariably, one of the goals of neuropsychological assessments is to be able to perform individual predictions: whether and when a conversion from an MCI to AD occurs. The data from meta-analyses indicate that deficits in episodic memory (i.e., the inability to remember new information) and executive functions (initiating, planning, and controlling activities) may be associated with a higher risk of AD [15]. Individuals without a progression of cognitive impairment scored significantly better than those with progression on the tests assessing divided and sustained attention, executive functions, immediate and delayed recall results, processing speed, visuospatial function, and working memory [16]. The verbal fluency and Trail Making (especially Part B) tests can be useful in screening assessments—studies have reported a meaningful decline as early as one year [17].

The studies indicate significant inter-individual variability in executive functions across the lifespan [18,19]. One study observed a significant variability in trajectories within the executive functions over a 40-year aging interval and varied patterns of decline in executive functions [20]. A consideration of the disease continuum brings to light the need to tailor both the diagnosis and therapy with intra- and inter-individual variabilities. Neuropsychology should therefore focus on improving the existing tests and developing new sensitive measures that map the changes in cognitive functions that reflect the neuropathological process specific to AD [21]. This can be achieved by taking advantage of the growing body of knowledge from cognitive neuroscience regarding the links between specific cognitive functions and the neuronal mechanisms that are susceptible to the earliest pathological changes [22].

On the basis of the presented data, it seems important to introduce neuropsychological diagnosis tests complementary to screening and aimed at parameters assessing the executive functions and psychomotor speed in the prodromal stages of dementia (dementia as a continuum) and mild cognitive impairment, as a complement to the assessment aimed at episodic memory and processes of remembering new information. In such an approach, the processes in question would constitute another neuropsychological marker. After reviewing the methods, we decided to verify the usefulness of the Comprehensive Trail Making Test in a group of geriatric ward patients. The Comprehensive Trail Making Test is chosen because it is an extension of the commonly used Trail Making Test (TMT) and, to date, its value has not been verified very often in Polish studies in relation to older patient populations. The purpose of our study is to determine its usefulness in diagnosing mild cognitive impairment. The purpose of our study is also to determine its psychometric parameters, which may be useful for other professionals involved in the diagnosis of cognitive functions in the older population. It should be noted that the TMT has been successfully used as a diagnostic tool for MCI and dementia. For this reason, we decided to conduct a preliminary validation of the CTMT, hoping that, as an expanded version of the TMT, it would add additional information to the diagnostic process to better assess an older patient’s cognitive functioning, providing additional quantitative and qualitative information on the mechanism of cognitive impairment. The results obtained may constitute pilot material.

## 2. Materials and Methods

### 2.1. Study Design, Setting, and Duration

The study was conducted on 98 patients hospitalized at the John Paul II Geriatric Hospital in Katowice (Poland). The patients were hospitalized for a comprehensive geriatric evaluation. The study was conducted between 2016 and 2018. A total of 106 patients were screened; 8 patients were excluded from the final analyses due to statistically significant differences in terms of age and education between the study groups. All participants that fulfilled the inclusion and exclusion criteria and agreed to participate were recruited for this study. Patients gave voluntary and informed consent to participate in the study. Patients were provided with blank informed consent forms and additional materials explaining the purpose of the study and how it was conducted. 

### 2.2. Sample Size and Participant Recruitment

In order to determine the appropriate sample size for the study, we conducted a power analysis, considering a significance level of α = 0.05, a desired power value of 0.90, and an expected effect (Cohen’s d) of 0.66. The result of the power analysis indicated that a sample size of at least 49 was required to obtain the desired power value. All the participants were assigned to 2 groups of 49 patients each by diagnosis:A group of patients with a diagnosis of MCI (Group I),A group of patients without a cognitive impairment, constituting the control group (Group K).

### 2.3. Study Inclusion, Exclusion, and Diagnostic Criteria

Patients over the age of 60 years were included in the study. Due to the discriminatory value of individual screening tests indicated in the literature, the MoCA scale proved to be the main diagnostic criterion in MCI diagnostics [14]. Patients who scored 19–25 on the MoCA scale were included in Group I, and those with an MoCA score of 26–30 were included in Group K. We adopted such score ranges based on previous studies conducted on the Polish older population [22]. Since the data indicate that combining screening tests increases their diagnostic accuracy in assigning patients to groups, the MMSE was also used (a score range of 24–26 for MCI, a score range of 27–30 was classified as correct; scores adjusted for age and education). In a situation where a patient scored below the cutoff point in only one of the screening tests (MMSE or MoCA), she/he was classified in the MCI group.

Patients in Group 1 in the screening cognitive assessment manifested a reduction in multiple cognitive domains, including short-term memory, sustained attention, and executive functions, which led to the diagnosis of an amnestic-type MCI with a concomitant impairment of other cognitive functions [9].

The factors excluding participation in the study were visual impairment (glaucoma and cataracts), diagnosed dementia, behavioral disorders, impairment of activities of daily living (simple activities of daily living, i.e., ADL < 3 points), history of stroke, history of traumatic brain injury, frailty syndrome, Parkinson’s syndrome, epilepsy, rheumatoid arthritis, cancer, symptomatic circulatory insufficiency, respiratory and renal diseases, diseases of the endocrine system (hyper- and hypothyroidism), uncompensated diabetes mellitus, mental disorders, including depression (also late-life depression—the Geriatric Depression Scale was used for exclusion), schizophrenia, alcohol dependence syndrome, use of sedatives, sleep medications, nootropics, precognitive drugs, and psychoactive substances.

### 2.4. Data Collection and Instruments Used

The initial examination of the patients admitted to the geriatric ward used the following set of tests: the MoCA (Montreal Cognitive Assessment), MMSE (Mini-Mental State Examination), CDT (Clock Drawing Test), and Geriatric Depression Scale (GDS). Another examination included an extended neuropsychological diagnosis, which included the Comprehensive Trail Making Test (CTMT) and verbal fluency test. There was a minimum gap of two days between the abovementioned two stages of the study. The variables that could affect the results obtained in the study were continuously monitored, including the level of severity of stress, the course of physical rehabilitation, and the somatic state of the patients. The first and second examinations were conducted by a psychologist in an office. In case of an error (such as combining incorrect points), the patient was informed about it by the psychologist. No maximum time limits were imposed; we were concerned with verifying that the patient was able to complete the test task. A brief description of the CTMT is given below.

Comprehensive Trail Making Test (CTMT)—a psychological test of the ability to focus attention on visual–spatial material—assesses visual search ability, psychomotor speed, as well as the ability to switch attention between stimuli of different types, which is one of the manifestations of working memory and executive functions. The CTMT was developed to extend the original TMT test, provided a more accurate assessment of cognitive functions, and was more comprehensive than the standard version of the TMT. The test was developed by Cecil Reynolds [23]. The direction of lines in the CTMT remains much more complex and varied than in the original TMT, requiring a greater performance of executive functions, including cognitive control and behavioral monitoring, the aforementioned psychomotor speed, and performance of visuospatial processes. The CTMT includes five trails that present varying levels of complexity and difficulty, and a variable number of distractors. In Trail 1, the patient drew a line to connect in order the numbers 1 through 25; each contained in a plain, black circle. In Trail 2, the patient drew a line to connect in order the numbers 1 through 25; each contained in a plain, black circle (29 empty distractor circles appeared on the same page). In Trail 3, the examinee drew a line to connect in order the numbers 1 through 25; 13 empty distractor circles and 19 distractor circles containing irrelevant line drawing appeared on the same page. In the following section, in Trail 4, the patient drew a line to connect in order the numbers 1 through 20, where 1 of the numbers were presented as Arabic numerals (e.g., 1 and 7) and the remaining numbers were spelled out in the English language form (e.g., “nine”). In Trail 4, there were circles with numbers and boxes with words; the words in the boxes were number words (for example, “nine”). The patient’s task was to draw a line from 1 to 2 and so on, connecting the circles to the boxes in the correct order. In Trail 5, the patient drew a line to connect in alternating sequences the numbers 1 through 13 and the letters A through L, beginning with 1 and drawing a line to A, then 2, then B, and so on until all the numbers and letters were connected (15 empty distractor circles appeared on the same page). The patient had to ignore all the circles where there were no letters or digits. For all the trails, the patient’s task was to connect the points as rapidly as possible. Trail 1 was similar to Part A of the TMT. Trail 5 in the CTMT was similar to part B of the TMT. Errors defined as marking a number or letter out of sequence were noted, but they were not converted to any form of standardized or scaled score. An error had a negative impact on the examinee’s score and these corrections added to the time needed to complete each trail.

Based on the statistical analysis conducted by the author of the test, two main factors of the CTMT were separated. The first included the so-called simple sequences—trails numbered 1, 2, and 3—, while the second included complex sequences: trails numbered 4 and 5 [23]. The reliability analysis conducted by the author of the test showed the following parameters of the individual trails: CTMT1 r = 0.98, CTMT2 r = 0.98, CTMT3 r = 0.96, CTMT4 = 0.98, and CTMT5 = 0.96. In the original version, there was an overall score, which was obtained from the converted results [23]. In all the tests, the indicator was the time of completion of the task. The time required to administer the CTMT ranged from approximately 5 to 12 min [23].

According to the author, the CTMT test can be performed on patients between the ages of 11 and 74 years, who can understand the directions for the subtest, who are able to formulate the necessary responses, and who can pass the practice items [23]. Although the patients who participated in our study were older (>74 years), they met all the requirements listed above in the test handbook to be eligible for the study.

In our study, the interpretation took into account the overall score, which was the sum of all the raw scores from each part, and the score from each trail separately. The interpretation was based on the raw scores due to their more frequent use in clinical practice and the availability of norms only for the American population [23]. Cognitive flexibility indices were also included in the calculation to eliminate the psychomotor component, which could be impaired in the course of normal physiological aging of the nervous system. The indicators are calculated as follows:$$Indicator1=\frac{CTMTTrail4+CTMTTrail5}{CTMTTrail1+CTMTTrail2+CTMTTrail3};$$
$$Indicator2=\frac{CTMTTrail5}{CTMTTrail1}.$$

Verbal fluency test—the patient is given 60 s to produce as many unique words as possible within a semantic category (e.g., animals—semantic fluency) or starting with a given letter (e.g., F, A, S—phonemic fluency). Executive functions, semantic retrieval, processing speed, and working memory are involved in verbal fluency tasks. The low score was a risk factor for MCI’s progression to dementia [24,25]. In our study, we used semantic fluency (category: animals) and phonetic fluency (words beginning with the letter “K”). The performance indicator was the number of words given by patient.

### 2.5. Statistical Analyses

The statistical analysis was performed using Python version 3.10. (Python is made available under the Python Software Foundation License (PSFL), which provides users with full rights to use the language, http://www.python.org). Descriptive analyses were presented for the demographic and cognitive data. The results were analyzed by groups. The Shapiro–Wilk test was used to assess the normality of the distribution of the parameters. Then, if possible, the *t*-test (or Welch’s test for unpaired samples with unequal variances) was used; otherwise, the Wilcoxon test was applied. Spearman’s rank correlation was used to assess the correlation coefficient. The statistical evaluation assumed a significance level of *p* < 0.05. Diagnostic performance was assessed using the area under the curve (AUC), sensitivity (Sen), specificity (Sp), and positive (PPV) and negative predictive (NPV) values. The receiver operating characteristic (ROC) curve was used to calculate the best cutoff point. The sample size was calculated in accordance to the logical justifications as proposed by Lakens [26].

## 3. Results

### 3.1. Characteristics of Study Sample

The study groups did not differ in terms of age and education level. The results obtained are presented in Table 1. The results of the screening tests are presented in Table 1. The frequency of the selected somatic diseases in the study groups is presented in Figure 1.

Statistically significant differences were found between Group I (MCI) and the control patients in terms of the CTMT completion time (*p* < 0.01). Patients with an MCI took longer to complete all the trails of the test, and statistically significant differences were obtained for the proposed cognitive flexibility indicators (Table 2). In order to assess the clinical relevance of the results of the comparative analysis between the group of patients with MCIs and the control group, we conducted an effect size analysis for the study variable. Cohen’s d, calculated from the analysis of the CTMT test (overall score) results between the mild cognitive impairment (MCI) patient group and control group, was 1.01 (95% confidence interval: 0.59–1.43), which meant that there were differences between the CTMT test results of the MCI patient group and control group. The Cohen’s d values for each CTMT trial suggest that patients with MCIs find it difficult to perform the tasks of each part of this test compared to the control group (Table 2). The profile of scores for each part of the CTMT in the group of patients with an MCI and in the control group are presented in Figure 2. The raw scores of the individual CTMT trails in the MCI patient and control groups are shown in Figure 2.

### 3.2. Diagnostic Accuracy and Optimal CTMT Cutoff Scores

In Trail 1, the cutoff point for patients with an MCI was 84 s with a sensitivity of 0.75 (CI: 0.61–0.85) and specificity of 0.63 (0.49–0.73). In Trail 2 for the MCI, the cutoff point of 99 s was associated with a sensitivity of 0.67 (CI: 0.53–0.78) and specificity of 0.81 (CI: 0.69–0.90). For Trail 3, the cutoff point was 104 s with a sensitivity of 0.63 (CI: 0.49–0.73) and specificity of 0.77 (CI: 0.64–0.86). In Trail 4, the cutoff point of 106 s for the MCI guaranteed a sensitivity of 0.73 (CI: 0.59–0.83) and specificity of 0.79 (CI: 0.66–0.88). In Trail 5, the cutoff point for MCI patients was associated with a sensitivity of 0.79 (0.66–0.88) and specificity of 0.81 (0.68–0.90). In an analysis of the effectiveness of the CTMT test in diagnosing cognitive impairment, the AUC results were mixed. In Trail 1, the AUC value was 0.69 (CI: 0.56–0.82), indicating the moderate ability of this part of the test to distinguish between the MCI patient and control groups. In Trail 2, the AUC value was 0.74 (CI: 0.62–0.86), suggesting that Trail 2 had a moderately effective diagnostic ability in distinguishing between the two groups. In Trail 3, an AUC of 0.70 (CI: 0.57–0.83) was achieved, indicating that this part of the test had a moderate ability to diagnose cognitive impairments. In Trail 4, the AUC value was 0.76 (CI: 0.64–0.86), suggesting that Trail 4 was relatively effective in diagnosing cognitive impairments in the MCI patient group. Trail 5 achieved the highest AUC value of 0.80 (CI: 0.69–0.91), indicating its great ability to distinguish between the MCI patient and control groups. The AUC value for the CTMT overall score was 0.77 (CI: 0.65–0.89), indicating the generally good ability of this test in identifying cognitive impairments in the MCI patient group. The CTMT overall score and CTMT 5 had the highest AUC scores (CTMT overall score = 0.77, CTMT Trail 5 = 0.80). The results obtained are presented in Figure 3 and Table 3.

The positive predictive value (PPV) and negative predictive value (NPV) helped determine how many cases of cognitive impairments the test correctly identified (PPV) and how many healthy individuals it correctly identified (NPV). The CTMT overall score achieved a PPV of 0.79; the NPV was 0.75. Trail 1 had a moderate PPV, which was 0.66; the NPV was 0.71. Compared to Trail 1, Trail 2 achieved a higher PPV of 0.77; the NPV was 0.71. For Trail 3, the PPV was 0.73; the NPV was 0.67. Trail 4 showed a PPV of 0.77 and the NPV was 0. 74. Trail 5 had a very high PPV of 0.80; the NPV was 0.79. The PPV and NPV results varied between the trails, which may indicate the varying ability of each trail to classify the scored correctly. Trail 5 seemed to show the greatest ability to classify the scores correctly, with very high PPVs and NPVs. The results obtained are presented in Table 3.

Regarding the screening tests, in Group I (MCI), the Spearman’s rank correlation showed a statistically significant negative correlation between the MoCA test and CTMT (overall score) (R = −0.28, *p* = 0.04), CTMT Trail 2 (R = −0.28, *p* = 0.04), CTMT Trail 3 (R = −0.37, *p* = 0.01), and CTMT Trail 5 (R = −0.35, *p* < 0.01). The lower the score on the MoCA test, the longer the time to complete each trail of the CTMT in the group of patients with MCIs. In Group I, there was a statistically significant negative correlation between the CDT test CTMT Trail 2 (R = −0.29, *p* = 0.04) and CTMT Trail 5 (R = −0.27, *p* = 0.05). The lower the CDT score, the longer the time it took to complete each trail for the CTMT. For the MMSE, no statistically significant correlations were obtained. The results are presented in Table 4.

In Group I (MCI), the Spearman’s rank correlation showed a statistically significant negative correlation between the semantic fluency and CTMT total score (R = −0.52, *p* < 0.01), CTMT Trail 1 (R = −0.47, *p* < 0.01), CTMT Trail 2 (R = −0.47, *p* < 0.01), CTMT Trail 3 (R = −0.51, *p* < 0.01), CTMT Trail 4 (R = −0.36, *p* = 0.01), and CTMT Trail 2 (R = −0.54, *p* < 0.05). A lower number of words mentioned in terms of semantic fluency in the group of patients with MCIs was associated with a longer CTMT performance. No statistically significant correlations were obtained for phonetic fluency (Table 5).

## 4. Discussion

The spectrum of cognitive deficits between normal physiological aging and dementia remains so wide that the need to assess the diagnostic value of individual tests used in neuropsychological diagnoses is increasingly emphasized. Tests assessing the performance of episodic memory and executive functions have been identified as the best individual indicators of future cases of dementia [15].

The CTMT used in the present study differentiated the study groups of patients with MCIs from those with preserved cognitive performances, which was consistent with the results obtained from other studies. It should be noted that the CTMT has not yet been validated in a group of older patients in the Polish population. In interpreting the results, we also referred to studies that used TMT to diagnose cognitive functions in the older patients. In the study by Ashendorf et al. [27], the TMT significantly differentiated between a group of patients with MCIs (*n* = 200) and a control group (*n* = 269). Patients with MCIs who scored better on the TMT Part B had higher scores on the MMSE. Kim et al. [28] also indicated that the TMT was an effective tool for differentiating the MCI from the control group. In a study by Zhou et al. [29], the TMT test differentiated the control group from patients with amnestic MCIs, while achieving high sensitivity and specificity parameters. In a group of 1051 cognitively normal subjects with a mean age of 67 years, the mean time of the TMT A was 39 s, the performance of part B was 89 s, and in both parts, a quicker performance was observed for women and the effect of the number of years of education on obtaining better results. A deterioration in the TMT performance with age was also observed.

Different cutoff points for different trails of the CTMT show varying characteristics. Some had a good ability to detect positive and negative cases, while others showed a moderate ability. The choice of an appropriate cutoff point depended on the clinical priorities, such as minimizing false negatives or false positives. The CTMT Trail 5 achieved the highest values for both sensitivity and specificity, and the highest AUC score (sensitivity: 0.79, specificity: 0.81, AUC: 0.80). The overall CTMT score also showed a good performance in distinguishing between positive and negative cases (sensitivity: 0.73, specificity: 0.81, AUC: 0.77). Trail 4 of the CTMT (sensitivity: 0.73, specificity: 0.79, AUC: 0.76) also had good results, with a particularly high specificity. Subsequent trails, such as CTMT 2 (sensitivity: 0.67, specificity: 0.81, AUC: 0.74) and CTMT 3 (sensitivity: 0.63, specificity: 0.77, AUC: 0.70), showed moderate efficacy results, with slightly lower sensitivity levels. The least effective trail appeared to be CTMT 1 (sensitivity: 0.75, specificity: 0.63, AUC: 0.69), which showed lower specificity compared to the other trails. This could lead to the conclusion that patients with an MCI experienced greater difficulties in the trails, which were more complex and whose completion required efficient executive functions (and not just attention and psychomotor speed). On the other hand, the CTMT functions as a whole. For that reason, the fact that it contains differentiated trails (easier and more difficult) testifies in favor of the tool. In a neuropsychological examination, a situation where a patient had trouble with every test task could be frustrating. In addition, the differentiation of individual trails created the possibility of plotting the performance profile of a given patient, which was proposed in the American version (for that reason, Figure 2 was proposed) [23]. The performance time in each part of the CTMT test increased in successive parts of the test (from Trails 1 to 5). Some of the CTMT patients showed a relatively good performance in Trails 1, 2, 3, and 4, and deterioration in Trail 5 (Figure 2), which may indicate executive dysfunction, with a simultaneously normal psychomotor speed and attentional processes. The group of patients with MCIs proved to be highly heterogeneous in the same parts of the test; the qualitative analysis allowed us to observe variabilities in the CTMT performance in the group of patients with MCIs (Figure 2). In contrast, even in the Trail 5, significant variations in the performances were observed. It was also observed that some patients tended to perform more slowly in all trails of the test than the rest of the group (Figure 2). On the other hand, an analysis of the raw scores indicated that some patients with MCIs scored well on the CTMT, achieving short execution times (below the cutoff point), which may suggest that other deficits (such as episodic memory) may have been prevalent in these patients. The results obtained using the CTMT indicate a high heterogeneity among patients with MCIs in terms of the severity of executive dysfunctions, which remains consistent with the data from the literature [18,19,20].

The results of the quantitative and qualitative analyses indicate that patients with MCIs take longer to complete tasks in all the CTMT trails, suggesting impaired psychomotor speed processes, sustained attention, working memory, and executive functions. A reduced capacity of the working memory, especially its central executive system, results in the impaired supervision of the entire integrated course of ongoing information processing, which leads to the rapid depletion of limited attentional resources. Therefore, the role of impaired control mechanisms (a parameter assessed by the TMT and CTMT, among others) is increasingly indicated among executive dysfunctions. Executive functions impairment is increasingly common in the early stages of Alzheimer’s dementia, manifesting itself especially in tasks involving control, cognitive flexibility, and response inhibition (dimensions of executive functions that can be evaluated by the CTMT) [30].

Increasingly, in indicators of the early stages of dementia, but also in cognitive impairments co-occurring in neuropsychiatric disorders, such as schizophrenia and bipolar affective disorder, the slowing down of one’s information processing speed is singled out as a dominant aspect across the spectrum of diverse cognitive dysfunctions [31,32]. Increasingly, in neuropsychology, cognitive functions are divided into “domain-specific” components with distinct functions and abilities (e.g., verbal and visuospatial) and “general domain” resources, such as processing speed. Neuropsychological studies of hierarchical cognitive organization indicate that the deficits observed in domain-specific tasks (e.g., impaired verbal memory) may be secondary to the general domain [33]. In a patient with an MCI, as indicated by the results we obtained using the CTMT, the problems in the working memory and difficulty of initiating an effective task performance may be due to the inability to implement a more automatic mode of information processing through the use of readily available associations and the rapid generation of stimulus-response relationships in the initial stages of the cognitive process [34]. Patients with MCIs during the CTMT had difficulties adequately processing the purpose of the task and inhibiting their automatic responses (e.g., performance involving connecting numbers to consecutive numbers instead of alternating letters). For this reason, Trail 1—“subjectively” the easiest—received the lowest AUC values. In the performance of more complex trails, patients with MCIs had to activate more controlled, effortful processing during the first stages of the task, slowly gaining proficiency in making associations between the stimulus and response (as especially observed in the CTMT Trail 5 results). It is likely that the connections of the prefrontal cortex with posterior areas and subcortical structures guarantee proper control, allowing the monitoring of intentional activities (an aspect of executive functions) [35]. The analyses conducted showed statistically significant correlations of the CTMT with the screening tests (MoCA) and the tests used in the neuropsychological assessment (verbal fluency test). All of the aforementioned tests were more focused on assessing the executive functions, such as working memory, which could be useful in the diagnosis of MCI [7,8,9].

Relating the obtained results to the diagnostic problems occurring in daily clinical practices and also to the prevalence of MCIs in the older population, for the CTMT (overall score), the PPV and NPV scores were at an acceptable level, suggesting that the test could identify well both cases of cognitive impairments and the lack of deficits. The results for Trails 1–5 are mixed; however, for the most part, the PPV and NPV values are at an acceptable level, meaning that individual trails can effectively identify cases of cognitive impairments, like the condition where the measured cognitive functions remain normal. The CTMT may be useful in this aspect for key diagnostic points in the neurodegenerative process (normal cognitive functions vs. mild cognitive impairments; mild cognitive impairments with a risk of conversion to mild dementia) [4,5]. The analysis of the CTMT results in the control group showed that the greatest variability in the results, with significant outliers, occurred in Trails 4 and 5. In the control group, Trails 1, 2, and 3 appeared to be more balanced, but still contained outliers that were significant observations (Figure 2). It was observed that for a small number of patients in the control group, the CTMT performance remained similar to that of patients in the MCI group, which may suggest a deficit in executive functions that was not identified in the first screening (MoCA and MMSE).

Based on the results, it can be suggested that the CTMT can be a useful test for evaluating patients with MCIs, but should not be the only tool used for diagnoses. Moreover, at the early stage of differentiating people with mild cognitive impairments from, the cognitively normal group, it is the speed of information processing that can be a valuable prodromal sign. At the same time, it should be emphasized that the timing of a task’s performance is more relevant for younger patients (<80 years), in whom we can expect better (faster) CTMT performances; in older patients (>80 years), effective task performances, regardless of the timing of the performance, is a more valuable indicator in the clinical evaluation.

In addition to the analysis of biomarkers, it was noted that one of the possible neuronal mechanisms associated with pathological state is a progressive deterioration in white matter. Lesions in the frontal white-matter area may affect executive functions in patients with AD [36]. One of the indicators for assessing white matter is the measurement of its integrity, which is based on determining clusters of white-matter hyperintensity (WMH) foci in the brain. A loss of white-matter integrity was a mediating factor between age and the speed of cognitive processing [37]. Studies have identified a correlation between the performance in the TMT-B (Trail Making Test Part B) and WMH burden [38]. The research data indicate that patients with WMHs take longer to perform the TMT than those in the control group [39]. There is strong evidence that WMH is associated with cognitive decline and an increased rate of progression of early onset as well as late-onset dementia [40,41]. Particularly sensitive to the changes in white matter are processes such as information processing speed, executive functions, and mnesthetic processes (including immediate and delayed recalls) [42,43]. WMHs are associated with cognitive impairments in the general population. A meta-analysis analyzed 12 cross-sectional studies of AD (total *n* = 1370, median age: 75) and also 10 studies of MCIs (nine cross-sectional, one longitudinal; total *n* = 2286, median age: 73) and found that the association between WMH and overall cognitive performance was significantly stronger for MCI than for AD. For both groups, the largest effect sizes were found for attention and executive functions and processing speed. No significant modifying effects of age or gender were found [44]. Presumably, the age-related changes in white matter, which WMH reflects, result in weakened connections between neurons and a reduced neuronal transmission speed, which, at the level of observed symptoms, manifests itself in the form of generalized slowing, significantly disrupting various cognitive processes and intensifying the physiological aging of the nervous system. It has also been increasingly advocated that white-matter changes should be considered as one of the verified markers of progression of cognitive impairment in Alzheimer’s dementia [45]. Some studies focus on the impact of WMHs on the connectivity of neuronal networks in the older population, indicating that disruptions within the frontoparietal and default mode networks are reflected in cognitive impairments, including executive dysfunction, processing speed, and attention [46,47]. The studies also indicate an association between WMH volume and frailty syndrome in patients with Alzheimer’s dementia [48]. WMH burden and APOE genotype clarify the relationship between blood pressure and cognitive functions, and may allow for a more accurate assessment of the impact of high blood pressure on cognitive decline and dementia risk [49].

It seems that psychological tests, such as the CTMT, require the involvement of multiple cognitive processes. Although, at the level of clinical interpretation, it poses numerous diagnostic difficulties (the need to identify the cognitive function that remains the most impaired and responsible for a poorer performance—a baseline defect in Luria’s terms). At the neuroanatomical level, by involving multiple areas of the brain, it creates an opportunity to capture structural and functional changes that are reflected in cognitive functioning (risk of false-negative errors, i.e., situations where the psychological test fails to identify a cognitive impairment in a patient in whom it occurs).

### Limitations

It is also important to note the limitations of this study. Despite the satisfactory results of the individual trails of the CTMT and the overall score, it should be pointed out that further empirical verifications on larger groups of patients with MCIs and dementia are needed to fully evaluate the potential of this test and to obtain better sensitivity and specificity values. The CTMT requires further analyses and verifications in more diverse clinical groups, including patients with Alzheimer’s dementia and its variants, vascular dementia, and frontotemporal dementia in the behavioral variant. In the future studies, it would be useful to estimate the diagnostic value of the CTMT in younger age groups, i.e., between 60 and 70 years old. In addition to the group of patients with dementia, future studies should also include patients with subjective cognitive decline. Another limitation of the present study was the lack of neuroimaging results in the form of CT or MRI scans in the study groups. Future studies would find it useful to determine the interrelationships in white-matter lesions and the CTMT results obtained for different age groups. In future studies, it would be useful to perform a full neuropsychological examination that would allow one to estimate the external validity of the CTMT (in our study, we only performed a verbal fluency test).

Another important aspect to verify the usefulness of the CTMT would be to monitor the studied group of patients over the years with regular follow-up sessions of cognitive functions in order to fully evaluate its usefulness in assessing the risk of conversion to dementia. Systematic and comprehensive documentations of the cognitive functions is important for monitoring the condition of a patient with a mild cognitive impairment. At present, the entire spectrum of variables is taken into account for the MCI, including cognitive status, biological markers, temporal dynamics of symptoms, and other (modifiable or not) risk factors for dementia [50]. In the following few years, we can expect the likely development of integrated models incorporating tests that assess cognitive functions, genetic risk factors, and the trajectory of change over time [51].

A well-established methodology for diagnostic test accuracy studies was described, including a phase that focused on the question of whether the test results could distinguish between people with the target disorder and those without it in real-world clinical situations. This type of research, called “pragmatic diagnostic test accuracy studies”, allows for a more realistic assessment of test performances in everyday clinical practices than the more controlled experimental studies. Pragmatic studies do not require strict inclusion and exclusion criteria, which makes it easier to recruit patients, and can be conducted in different clinics, increasing the external validity [52]. Our study was a pilot study and therefore numerous exclusion criteria were included; however, these may raise questions about the conclusions. Numerous exclusion criteria resulted in the collection of a small group of patients (*n* = 106), despite the relatively long duration of the study (2016–2018). Further studies of the usefulness of the CTMT without such strict exclusion criteria may be useful.

## 5. Conclusions

The Comprehensive Trail Making Test may be a useful tool in diagnosing mild cognitive impairments in older patients as a complement to screening tests.

## Figures and Tables

**Figure 1 geriatrics-08-00108-f001:**
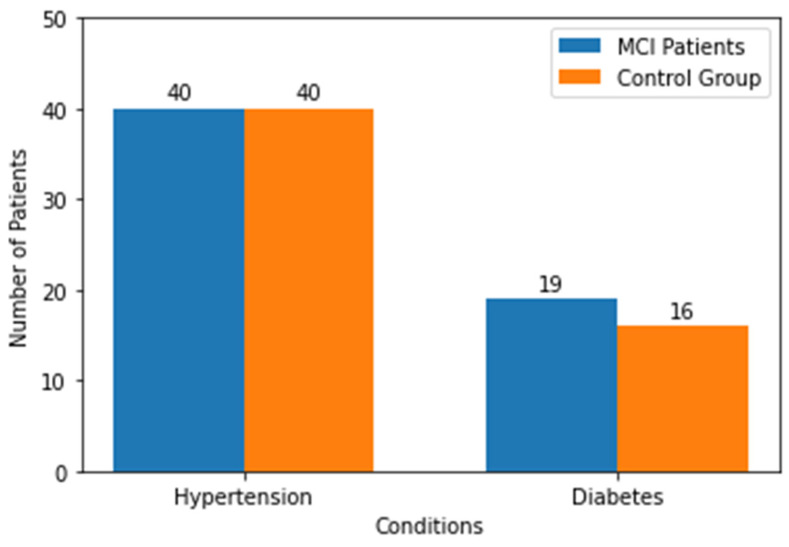
The frequency of selected somatic diseases in the study.

**Figure 2 geriatrics-08-00108-f002:**
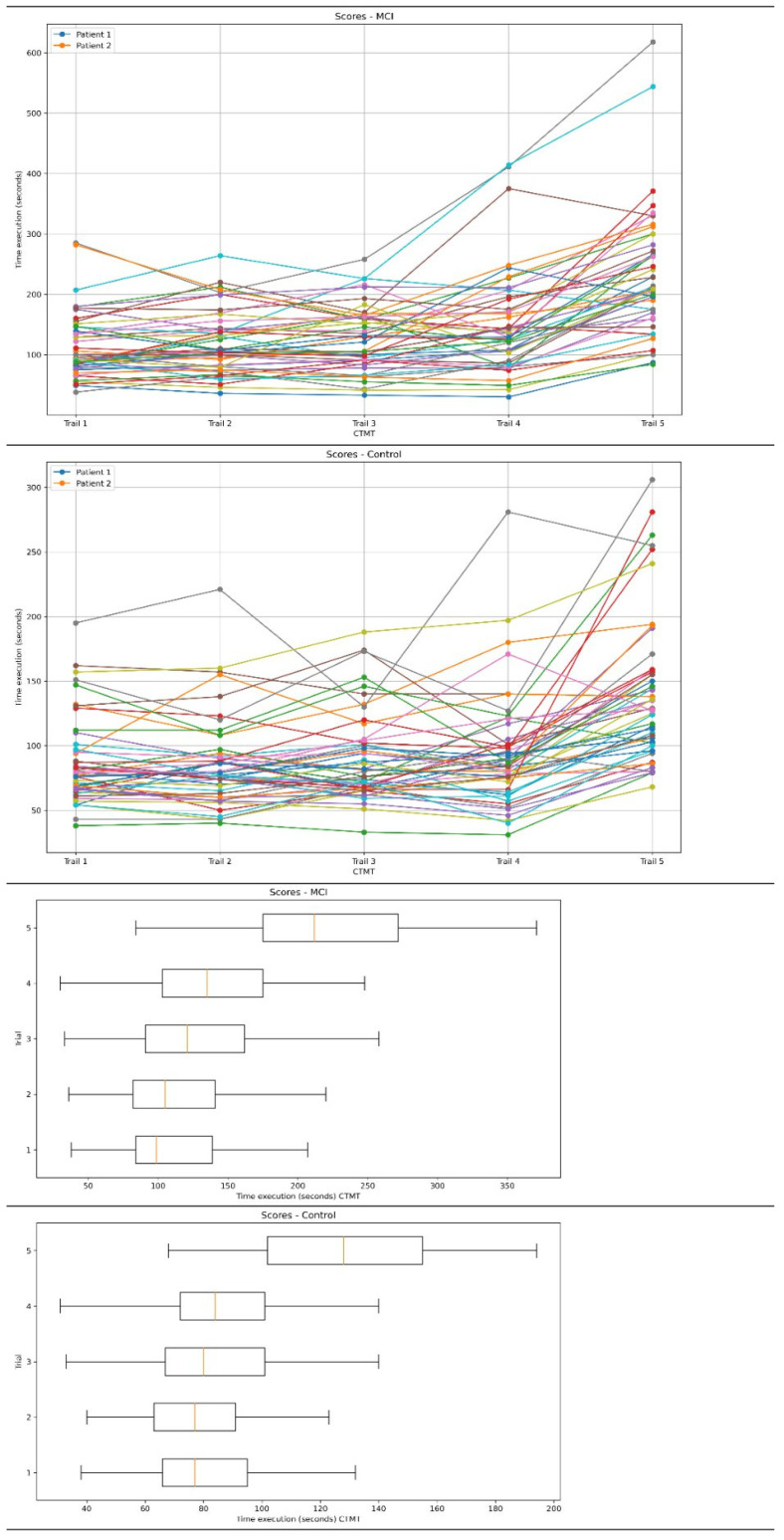
Profile of scores for each part of the CTMT in the group of patients with an MCI and in the control group. Raw scores for each part of the CTMT in the group of patients with an MCI and in the control group.

**Figure 3 geriatrics-08-00108-f003:**
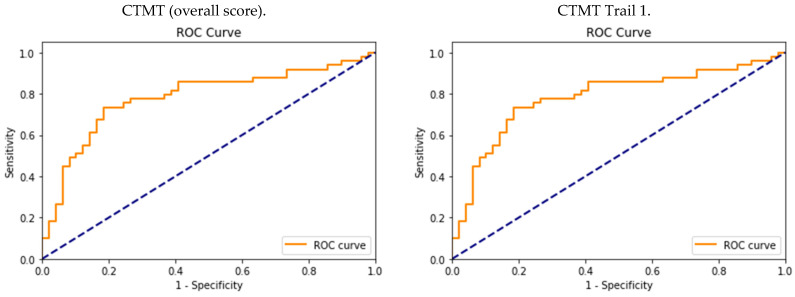
Cutoff points for the CTMT in the studied group of patients.

**Table 1 geriatrics-08-00108-t001:** The sociodemographic variables and screening test for patients with a mild cognitive impairment (Group I) and without a cognitive impairment (Group K) (M ± SD).

Variables	Groups		*p*-Value
	Group I (*n* = 49)	Group K (*n* = 49)	
Sex (female/male)	43/6	44/5	
Age (years)	77.24 ± 5.43	75.51 ± 6.11	0.07
Education (years)	10.20 ± 2.30	11.04 ± 2.44	0.08
MMSE	28.28 ± 1.48	29.06 ± 1.19	
MoCA	22.42 ± 2.02	27.95 ± 1.25	
CDT	8.86 ± 1.85	9.79 ± 0.57	
Semantic fluency	13.42 ± 4.49	18.4 ± 5.85	
Phonetic fluency	10.53 ± 4.07	13.29 ± 0.59	

MMSE = Mini-Mental State Examination, MoCA = Montreal Cognitive Assessment, CDT = Clock Drawing Test, *p* = statistical significance (*p* < 0.05).

**Table 2 geriatrics-08-00108-t002:** CTMT values for patients with mild cognitive impairments (Group I) and without cognitive impairments (Group K) (M ± SD).

Test		Groups		*p*-Value	Cohen’s d Value (CI)
		Group I (*n* = 49)	Group K (*n* = 49)		
CTMT (seconds)	All trails	744.44 ± 309.47	491.57 ± 169.79	*p* < 0.01	1.01 (0.59–1.43)
	Trail 1	114.16 ± 52.60	85.83 ± 31.98	*p* < 0.01	0.65 (0.24–1.05)
	Trail 2	119.59 ± 51.06	85.14 ± 34.46	*p* < 0.01	0.79 (0.37–1.20)
	Trail 3	125.89 ± 52.60	89.16 ± 33.26	*p* < 0.01	0.83 (0.42–1.27)
	Trail 4	150.93 ± 83.11	93.16 ± 43.82	*p* < 0.01	0.86 (0.45–1.28)
	Trail 5	233.85 ± 102.35	138.26 ± 54.09	*p* < 0.01	1.16 (0.73–1.59)
	Indicator 1	1.08 ± 0.23	0.90 ± 0.21	*p* < 0.01	---
	Indicator 2	2.13 ± 0.57	1.68 ± 0.62	*p* < 0.01	----

**Table 3 geriatrics-08-00108-t003:** Cutoff points for the CTMT in the patient groups studied.

	Diagnostic Performance					
	Cutoff	Sen	Sp	PPV	NPV	AUC
CTMT (overall score)	578	0.73 (CI: 0.59–0.85)	0.81 (CI: 0.68–0.90)	0.79	0.75	0.77 (CI: 0.65–0.89)
CTMT Trail 1	84	0.75 (CI: 0.61–0.85)	0.63 (CI: 0.49–0.73)	0.66	0.71	0.69 (CI: 0.56–0.82)
CTMT Trail 2	99	0.67 (CI: 0.53–0.78)	0.81 (CI: 0.68–0.9)	0.77	0.71	0.74 (CI: 0.62–0.86)
CTMT Trail 3	104	0.63 (CI: 0.49–0.73)	0.77 (CI: 0.64–0.86)	0.73	0.67	0.70 (CI: 0.57–0.83)
CTMT Trail 4	106	0.73 (CI: 0.59–0.83)	0.79 (CI: 0.66–0.88)	0.77	0.74	0.76 (CI: 0.64–0.86)
CTMT Trail 5	161	0.79 (CI: 0.66–0.88)	0.81 (CI: 0.68–0.90)	0.80	0.79	0.80 (CI: 0.69–0.91)

PPV—Positive Predictive Value; NPV—Negative Predictive Value; AUC—Area Under the Curve. Sensitivity (Sen); specificity (Sp); discriminatory capacity (diagnostic accuracy of the AUC) are reported with values (%) with corresponding 95% confidence intervals (95% CIs).

**Table 4 geriatrics-08-00108-t004:** Correlation coefficient between screening tests and CTMT in a group of patients with MCI (*n* = 49).

Test	MoCA		MMSE		CDT	
	R	*p*	R	*p*	R	*p*
CTMT (all)	−0.28	0.04	0.04	0.74	−0.23	0.11
CTMT Trail 1	−0.22	0.12	0.14	0.33	−0.22	0.12
CTMT Trail 2	−0.28	0.04	0.05	0.7	−0.29	0.04
CTMT Trail 3	−0.16	0.26	0.15	0.3	−0.35	0.01
CTMT Trail 4	−0.20	0.15	0.07	0.61	−0.13	0.34
CTMT Trail 5	−0.36	0.009	0.05	0.70	−0.27	0.05

R = Spearman’s correlation coefficient; *p* = statistical significance (*p* < 0.05).

**Table 5 geriatrics-08-00108-t005:** Correlation coefficient between the Verbal Fluency Test and CTMT in a group of patients with MCI (*n* = 49).

Test	Semantic Fluency		Phonetic Fluency	
	R	*p*	R	*p*
CTMT (all)	−0.52	<0.01	−0.17	0.23
CTMT Trail 1	−0.47	<0.01	−0.22	0.12
CTMT Trail 2	−0.47	<0.01	−0.12	−0.12
CTMT Trail 3	−0.51	<0.01	−0.04	0.74
CTMT Trail 4	−0.36	0.01	−0.24	0.09
CTMT Trail 5	−0.54	<0.01	−0.22	0.11

R = Spearman’s correlation coefficient; *p* = statistical significance (*p* < 0.05).

## Data Availability

Quantitative data are available upon request.

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
