# Peer review of "Utility of the Comprehensive Trail Making Test in the Assessment of Mild Cognitive Impairment in Older Patients"

_geriatrics, 2023, doi:10.3390/geriatrics8060108_

Round 1

Reviewer 1 Report

Comments and Suggestions for Authors

Review: Usefulness of the Comprehensive Trail Making Test in the diagnosis of mild cognitive impairment in older patients.

 The manuscript describes the application of the Comprehensive ail Making Test (CTMT) in a group of hospitalized patents older than 60 years, comparing those with a diagnosed mild cognitive impairment(MCI) versus those without MCI.

 The study has an interesting and important topic and seems well-executed. There are some points, though, which warrant attention.

 Introduction

The statements in the initial part of paragraph 2 (from “Modern diagnostic criteria for MCI focus on expanding the cognitive disorder classification scheme…” to “… AD presentation with frontal lobe and executive function impairment)” need references.

 The authors could perhaps check how to make two or three paragraphs out of this long paragraph.

 The Introduction should be finished with a short outline of which data they will collect, what they will learn from the data, and what they expect to achieve by this. They later state “The purpose of the present study was to determine the usefulness of the Comprehensive Trail Making Test in diagnosing mild cognitive impairment in older patient.” They should already say this here, and they should also announce how they will determine this.

 Methods:

 A description is missing of why participants were hospitalized. What were the reasons for intake, what were the diagnoses (except those listed at exclusion criteria), which treatments did they undergo?

 How many patients were screened, and how many were eligible and declined participation?

 There is a very long list of exclusion criteria. How does this affect conclusions and the practical use of the CTMT?

 He Comprehensive Trail Making Test (CTMT) is described, but what are the different parts meant to measure? Here are also no references. Who did develop this instrument, and how was validity in the past?

 The concrete testing procedure should be described. Who carried out the tests together with the patients, in which setting, how was it handled if errors (particularly, wrong connections) occurred, were there maximum time limits, and how many participants did reach these limits?

 A description is missing of which statistical analyses are planned.

Results

 “The Shapiro-Wilk test was used to assess the normality of the distribution of the parameters. Then, if possible, the t-test (or Welch's test for unpaired samples with unequal variances) was used, otherwise the Mann-Whitney test was applied.” These issues should rather be included in the Methods section. Moreover, the t-test does not require that the dependent variable be distributed normally in the total population. Instead, it is assumed that both populations separately are distributed normally. But this assumption is only required in small populations. With n(total) > 30, the central limit theorem tells us that the t-test (or Welch test, by default) can be applied regardless of the distributions. Therefore, you should simply use the Welch test for all comparisons.

 “Study groups did not differ” should rather be “did not differ statistically significantly”. The average age of group I was 2 year (or 1/3 SD) lower that in group K, and average years of education was 1 year lower. Both are characteristics, which according to the authors affect results. The authors should investigate to which degree group differences in terms of mean scores change if they adjust for age and education.

 Table 2: Results should be also displayed graphically, preferably as boxplots. If there is not enough space then at least as supplementary information.

 “A cutoff point of 540 seconds was proposed for the Comprehensive Trail Making Test in the total score: “ It should be explicitly stated (in either the Methods section or here in the results section) how cut-offs were determined.

The 95% confidence intervals for sensitivity and specificity values should be given.

 Discussion:

 The authors should point out more clearly why there is a need for using the CTM instead of (or in addition to?) the TMT, given that TM has long been successfully used as a diagnostic tool in the context of dementia and MCI.

 The specifity and sensitivity of the suggested cutoffs should be characterized. Are these key figures satisfactory or maybe poor?

 What would these figures mean in practice, i.e. how would NPP and PPP look for some typical prevalence rates, in typical environments of dementia/MCI testing?Dea Editors,

Reviewer 2 Report

Comments and Suggestions for Authors

The subject of the ms is interesting. Thank you very much for considering me a potential reviewer and giving me the chance to read it.

The main issues that I detected have to do with the methodology of the study. The authors divided the two groups of the sample into MCI and healthy participants mainly or only(?) on the basis of their MoCA scores and without taking into account a comprehensive neuropsychological assessment and/or neurological and neuroimaging examination. Moreover, the MoCA range for considering people as healthy or MCI does not seem to have a relative reference to a MoCA adaptation in the population of interest for this study. Is really a score of 19 indicative of MCI pathology and not of dementia? Is a score of 26 indicative of a cognitively healthy performance or of Subjective Cognitive Decline? In fact it seems that SCD was not tooked into account at all! As regards the healthy group, did the suffer from cardiovascular risk factors such as hypertension? These factors -in fact, the authors discuss the issue extensively in the discussion section- can significantly contribute to a lower performance in executive function tasks...

Moreover, there is not any information as regards the way the authors defined the N of the sample. Power analysis should be added in the methods section to ensure that the N was appropriate. 

As regards the results of the study, I would prefer to see a roc curve analysis with all its indices (AUC, P...) in a table. In fact, it seems that none of the indices of sensitivity and specificity of the test and its conditions is really important.

In the Discussion section, the authors should focus on the confirmation or not of the main aim of the study. They discuss more general and they do not focus on the findings of this study.                 

Comments on the Quality of English Language

only minor revisions are needed

Reviewer 3 Report

Comments and Suggestions for Authors

This study aims at determining the usefulness of a variant of the TMT for the diagnosis of MCI. The results are important and underline the importance of executive tests such as the CTMT for the diagnosis of MCI.

The main problem with the manuscript is the insufficient description of the CTMT. This description of the CTMT is unclear and insufficient and should be much more detailed. What is done in parts 1, 2, and 3? What is meant by taking turns? The formula at the end is unclear.
A figure with the different parts would be helpful and important.

Minor points:

why were scores in MMSE and CDT not additionnally used to assign the patients into groups I or K?

Table 1 could be omitted. The lab tests are not relevant. Age and education is certainly relevant and the results could be simply given in the text.

Comments on the Quality of English Language

the language quality is more or less o.k.

Round 2

Reviewer 2 Report

Comments and Suggestions for Authors

Given the methodological restrictions mentioned and the authors' additions-explanations in the text, I think that the ms. could be published as it is.